# Application of the Stripping Voltammetry Method for the Determination of Copper and Lead Hyperaccumulation Potential in *Lunaria annua* L.

Maša Buljac [1,*] , Josip Radić [1] , Marijo Buzuk [2] , Ivana Škugor Rončević [2] , Nives Vladislavić [2] , Denis Krivić [3] and Ana Marijanović [4]

1 Department of Environmental Chemistry, Faculty of Chemistry and Technology, University of Split, 21000 Split, Croatia; jradic@ktf-split.hr
2 Department of General and Inorganic Chemistry, Faculty of Chemistry and Technology, University of Split, 21000 Split, Croatia; buzuk@ktf-split.hr (M.B.); skugor@ktf-split.hr (I.Š.R.); nives@ktf-split.hr (N.V.)
3 Division of Biophysics, Gottfried Schatz Research Center, Medical University of Graz, 8010 Graz, Austria; denis.krivic@medunigraz.at
4 Faculty of Chemistry and Technology, University of Split, 21000 Split, Croatia; anamarijanovic@ymail.com
* Correspondence: masa@ktf-split.hr; Tel.: +385-21-329-479

**Abstract:** Various species of the Brassicaceae family are known to hyperaccumulate metals. *Lunaria annua* L., a plant from the Brassicaceae family, is an oilseed crop known for its pharmaceutical and nutraceutical applications. In this work, *Lunaria annua* L. was investigated for its accumulation potential in copper and lead-contaminated soil. Concentrations of copper and lead were measured before planting (in seeds and soils) and after the plant was harvested (in soils and plant). Two types of soils were used: a soil sample collected from the Botanical Garden of the Faculty of Science, University of Split (soil 1, S1) and a commercially available organic mineral substrate (soil 2, S2). Measured pH values showed that the S1 (pH = 8.58) was moderately alkaline soil. On the other hand, the purchased organic soil, S2 (pH = 6.35), was poorly acidic to neutral. For the determination of copper (Cu) and lead (Pb), square wave anodic stripping voltammetry (SWASV), using a glassy carbon electrode modified with mercury film, was applied. The concentrations of Pb and Cu were determined and calculated in the sample using the standard addition method. Obtained results have shown that *Lunaria annua* L. is a lead hyperaccumulator (4116.2 mg/kg in S1 and 3314.7 mg/kg in S2) and a potential copper accumulator (624.2 mg/kg in S1 and 498.9 mg/kg in S2). Likewise, the results have shown that the higher the pH is, the lower the possibility that metal accumulation exists.

**Keywords:** *Lunaria annua* L.; accumulation; lead; copper; square wave anodic stripping voltammetry; standard addition method

## 1. Introduction

Heavy metals are pollutants that significantly affect the environment and their toxicity has been a problem of increasing significance for ecological, evolutionary, nutritional and environmental reasons [1]. Due to human activity and consequent environmental pollution, heavy metals enter the biosphere, where, due to their toxicity, they lead to certain changes in specific metabolic pathways in living organisms. In addition to the organisms, heavy metals reach the soil. Technological operations that are a part of everyday industrial workflows are the main cause of soil contamination by deposition of harmful substances. Further, agricultural processes and waste disposal sites, as well as air pollution and pollutant deposition by wastewater treatment, account for a large portion of soil contamination causative factors [2]. Interestingly, growing plants on soils that contained increased levels of heavy metals led to the discovery of the metal-accumulating properties of many species known today. However, not all heavy metals have a toxic effect on plants. In other words, normal plant growth requires the presence of heavy metals, which is why it up to the type

of metal, its chemical form, pH, the composition of soil or sole plant species whether a metal will have deleterious effects on a particular plant or not [1].

The majority of metals cannot be eliminated from the environment by chemical or biological transformation. They cause a state of oxidative stress in the plant and it is necessary to better understand the mechanisms of their harmful effects, as well as the tolerance of certain species and the specific reaction of their varieties to the increased availability of heavy metals in the environment.

Lead, as a non-essential trace metal, is a strong environmental pollutant that is toxic in very low concentrations and accumulates in various parts of the plant. Copper is a micronutrient highly important for plants because of its ability to maintain vital plant activity. Any deficiency of a nutrient affects plant growth and decreases crop yields [3]. There are different sources of lead and copper in the environment, such as natural sources (geologic parent material or sedimentary rock, volcanoes, marine sources) and industrial sources (energy supplying power stations, metallurgy and electroplating, chemical industry, pharmaceuticals). Other sources of heavy metals include waste incineration, landfills and transportation (cars, diesel-powered vehicles and aircraft) [1]. For copper to be biologically available, the pH should optimally be between 4.5 and 6, while the mobility of lead does not depend on the pH value of the soil and is quite small due to its tendency to bind to the organic matter [4].

Crop plants have been used to extract heavy metals from soil and sediments, followed by translocation of contaminants to the harvestable stalks and leaves of the plants [2]. As many metals that are being stockpiled by plants are essential nutrients, there is great potential for their use in food fortification and phytoremediation. Thus, analysis of metal accumulation capacity represents a promising aspect of plant use [5]. Over 700 metal-hyperaccumulating and tolerant plant species are known, particularly in the context of affinity towards Nickel (over 500) [6]. Species associated with storing cobalt, copper or zinc in higher amounts are second in rank, although much smaller in number, with those hyperaccumulating arsenic, cadmium, gold, lead, manganese and thallium being in the third place [7].

According to Baker and Brooks [8], the largest numbers of hyperaccumulating species belong to the Brassicaceae family. Likewise, Sarma emphasizes in his paper [9] that metal hyperaccumulation is a property widespread among the representatives of the Brassicaceae family. Ni and Zn hyperaccumulation, the former being first discovered in 1948 in Alyssum bertolonii/Brassicaceae, and later in 1865 in Noccaea caerulescens (formerly, Thlaspi caerulescens)/Brassicaceae, began to attract increasing attention in the early 1990s as incidences of the alternative metal accumulating strategies [10].

A hyperaccumulator has been defined as a plant that can accumulate cadmium (>100 mg/kg), copper and lead (>1000 mg/kg), zinc (>10.000 mg/kg) in its shoot dry matter. Further, in these plants, metal concentrations in shoots are greater than in roots, showing a peculiar ability of a plant to absorb and transport metals, and store them in their above-ground parts [8]. A plethora of factors affect hyperaccumulation, and some of them are described in the work of Peng et al. [6].

Brassicaceae plants often feature regularly in diets as raw or preserved vegetables and vegetable oil [11].

*Lunaria annua* L. is a biennial cruciferous oilseed crop. The biennial character of *Lunaria* is the main constraint for economically feasible cultivation. The seeds contain 30–35% oil, which contains 67% of long-chain fatty acids (44% erucic acid, C22:1, and 23% nervonic acid, C24:1). The oil is suitable as a lubricant [12,13]. In representatives of the genus, the boiled root of a plant is edible, while unripe fruits can be chopped and used as a spice.

Determinations of heavy metals in environmental samples, whether qualitative or quantitative, can be performed using different spectroscopic and electroanalytical methods. These include atomic absorption spectrometry (AAS), inductively coupled plasma atomic emission spectroscopy (ICP-AES) [14], microwave-induced plasma optical emission spectroscopy (MIP-OES), inductively coupled plasma optical emission (ICP-OES) and

stripping voltammetry [15]. The latter is a widely used electrochemical technique for the detection of heavy metals in soil and water samples due to its ability to measure metal ions at trace concentrations. Stripping analysis consisted of two steps, a preconcentration and a stripping one. During the former one, the metal ions were reduced to a metal of interest and accumulated on the surface of a working electrode; during the latter, by applying a positive (anodic) or negative (cathodic) potential scan to the electrode, a metal of interest was oxidized back to its ion form and stripped out into the solution. In the meantime, the amount of the element was determined by measuring the generated current [3].

In this work, we investigated the accumulation properties of *Lunaria annua* L. in the context of copper and lead-abundant soil. Metal concentrations of copper and lead were measured before planting and after the plants were harvested in soils, seeds and plants depending on the stage of growth. COMPO SANA, a commercially available organic mineral substrate and a soil sample collected from the Botanical Garden of the Faculty of Science, University of Split (30 cm-deep soil surface), were used in these experiments.

For the determination of Cu and Pb, square wave anodic stripping voltammetry (SWASV), using a glassy carbon electrode modified with mercury film, was applied. The standard addition method was used to determine and calculate the concentrations of copper and lead.

## 2. Materials and Methods

### 2.1. Chemicals and Materials

All chemicals and reagents were at least of analytical grade, and Milli-Q water was used throughout the experiment. Lead(II) nitrate, potassium nitrate and phosphoric acid were purchased from T.T.T. (Zagreb, Croatia); copper(II) nitrate, mercury(II) nitrate monohydrate and sulfuric acid were purchased from Kemika (Zagreb, Croatia). Nitric acid was purchased from Merck (Darmstadt, Germany). Hydrochloric acid was purchased from VWR Chemicals (Radnor, Pennsylvania, SAD). Ethanol was purchased from Gram-mol (Zagreb, Croatia). Standard solutions ($1 \times 10^{-5}$ M) of each metal were prepared by dissolving exact salt weight ($Pb(NO_3)_2$), $Cu(NO_3)_2 \times 3H_2O$) in 100 mL deionized water.

All experiments were carried out in a conventional three-electrode electrochemical cell at 25 °C. The glassy carbon ($\phi$ = 3 mm) served as a working electrode, Ag/AgCl/3 M KCl as a reference electrode and platinum as an auxiliary electrode. Electrochemical measurements were carried out using a potentiostat (Autolab PGSTAT 302N), connected to a PC and driven by the GPES4.9 software (Eco Chemie).

The supporting electrolyte required for the determination of Pb(II) and Cu(II) was a mixture of 0.01 M $H_3PO_4$ and 0.01 M $HNO_3$ acids. To eliminate the influence of matrix in soil samples, measurements were performed with the addition of 100 µL aqua regia (2 mL of concentrated $HNO_3$ and 6 mL of concentrated HCl into a 50 mL volumetric flask and filled with deionized water up to the volume). All experiments were carried out at room temperature (approximately 25 °C) without removing the dissolved gases.

### 2.2. Sample Preparation

2.2.1. Preparation of Soil Samples

The soil sample collected from the Botanical Garden (hereinafter S1) and a commercially available organic mineral substrate (COMPO SANA) (hereinafter S2) were air-dried for three weeks, mixed into a homogenous mixture and then sieved through a 2.5 mm, 1.0 mm, 0.5 mm and 0.25 mm mesh.

In total, 1.0 g of each soil sample was put on porcelain crucibles, placed in a cool muffle furnace and ashed at 550 °C for 3 h. Afterwards, the ashes were cooled and dissolved in 8 mL of aqua regia. The solution was filtered, the ashes were washed with diluted aqua regia and the filtrate was transferred into a 50 mL volumetric flask and diluted to the mark with deionized water.

For soil pH determination, two replicates of a 10.00 g portion of each soil sample were individually transferred to 50 mL glass beakers, and 25 mL of Milli-Q water ($\Omega$ = 18.2 M$\Omega$/cm)

was added. A premium hotplate stirrer model MSH-20A (Witeg Labortechnik, GmbH, Wertheim, Germany) was used for stirring the solutions. The pH of the samples was measured using a pH/conductivity combimeter (Orion Star Series Meter Thermo Fischer Scientifc Inc., Beverly, MA, USA).

### 2.2.2. Preparation of Seeds Samples

In total, 1.0 g of the plant was put on porcelain crucibles, placed in a cool muffle furnace and ashed at 550 °C for 3 h. Afterwards, the ashes were cooled and dissolved in 8 mL of aqua regia. The solution was filtered, the ashes were washed with diluted aqua regia and the filtrate was transferred into a 50 mL volumetric flask and diluted to the mark with deionized water.

### 2.2.3. Preparation of Plant Samples

Four flowerpots were prepared for each plant species; two of them were filled with S1, and two of them with S2. The seeds of *Lunaria annua* L. were planted in the flowerpots with the prepared soils. Two flower pots were watered with 15 mL of tap water of pH 7 (samples S1 + $H_2O$ and S2 + $H_2O$) and the other two with 15 mL of a prepared solution which contained 8 mM of $Pb(NO_3)_2$ and 8 mM $Cu(NO_3)_2 \times 3H_2O$ (samples S1 + Pb/Cu and S2 + Pb/Cu), for thirty days. Unseeded and untreated soils were used as controls.

After thirty days of cultivation, when plants had flowered, a plant from each flowerpot was harvested and left to dry at room temperature (ca. 25 °C) over a period of two days, and afterwards, in a laboratory drying oven at 75 °C to a constant mass, in order to obtain dry mass. Plant samples were cut to pieces and milled into a homogenous powder. In total, 1.0 g of a plant was put in a porcelain crucible, placed in a cool muffle furnace and ashed at 550 °C for 3 h. Afterwards, the ashes were cooled and dissolved in 8 mL of aqua regia. The solution was filtered, the ashes were washed with diluted aqua regia and the filtrate was transferred into a 50 mL volumetric flask and diluted to the mark with deionized water.

### 2.3. Electrode Preparation

A working electrode was polished with alumina powder to obtain a mirror-like surface, washed with deionized water, sonicated in ethanol solution for 2 min, washed with deionized water and dried. After, the electrode was electrochemically cleaned in 0.5 M sulfuric acid in the potential area of −1 V to 1 V, and the scan rate of 0.2 $Vs^{-1}$.

Plating was carried out by immersion in a solution of 0.1 M $KNO_3$, 0.01 M $HNO_3$, and $2 \times 10^{-4}$ M $Hg(NO_3)_2 \times H_2O$, while the electrode potential was held at −1.0 V for 2 min. After each experiment, the electrode was electrochemically re-prepared, and the mercury film was cleaned by wiping the electrode with a wet tissue, followed by plating a new mercury film for performing a new experiment.

### 2.4. Voltammetric Measurements

The concentrations of lead and copper were determined using the square wave anodic stripping voltammetry (SWASV) under optimized parameters, i.e., electrodeposition potential and final potential of −1.5 and +0.25 V, respectively; electrodeposition time of 60 s; step potential = 2 mV, step amplitude = 25 mV, and frequency = 10 Hz.

Before the measurements, to avoid any contamination, the electrochemical cell was rinsed with concentrated $HNO_3$ and Milli-Q water.

### 2.5. Analysis of Samples

In total, 50 mL of supporting electrolyte was transferred into an electrochemical cell, 100 μL of each digested soil or plant sample and 100 μL of aqua regia was added. The solution was stirred thoroughly and the potential was scanned. The obtained values of the peak current are given in the results chapter and were used to determine the concentrations of Pb and Cu in samples, respectively.

### 2.6. Standard Addition Method

The standard addition method was used to determine the concentrations of metals in samples and to eliminate the matrix effects. In total, 100 μL of a prepared sample and a particular volume of standard solutions ($1 \times 10^{-5}$ M) of each metal were transferred into a 50 mL volumetric flask. The peak current value, relevant to each addition, was plotted on the *y*-axis, while the *x*-axis was graduated in terms of the concentration of a standard in the cell. The regression line was calculated and extrapolated back to the point on the *x*-axis at which *y* = 0. The content of the determined element in the sample (w) was calculated according to the following equation:

$$w \text{ (mg kg}^{-1}) = -c_x \text{ (mol L}^{-1}) \times M \text{ (metal/gmol}^{-1}) \times 2.5 \times 10^7 \tag{1}$$

where $2.5 \times 10^7$ represents the correlation factor.

## 3. Results

### 3.1. Simultaneous Determination of Cu(II) and Pb(II)

Since certain concentrations of lead and copper can be found in real samples at the same time, combinations were developed with constant $Pb^{2+}$ ($1 \times 10^{-7}$ M) with different additions of $Cu^{2+}$ ($1 \times 10^{-6}$; $1 \times 10^{-7}$; $1 \times 10^{-8}$ M) and constant $Cu^{2+}$ ($1 \times 10^{-5}$ M) with different additions of $Pb^{2+}$ ($1 \times 10^{-7}$; $1 \times 10^{-8}$; $1 \times 10^{-9}$ M).

The reductive peak currents for the Cu and Pb ions increased linearly with an increase in their respective concentrations without affecting the other peak currents (Figure 1). It is obvious that copper and lead do not interfere with each other. The reduction peak potentials of the Cu and Pb ions on the modified electrode were separated completely into two well-defined peaks at 0.00 V and −0.44 V vs. Ag/AgCl, respectively.

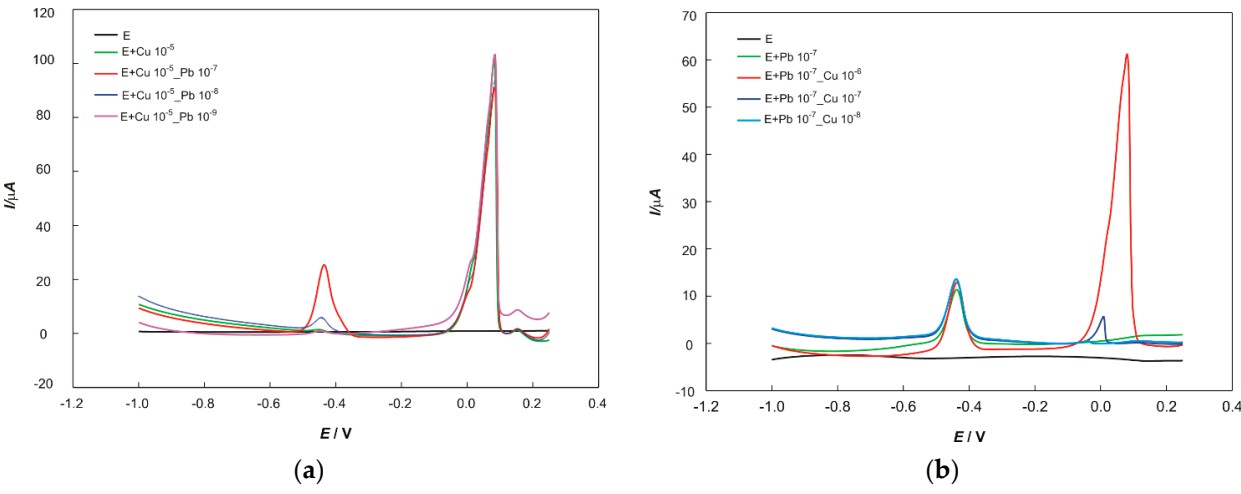

**Figure 1.** Square wave anodic stripping voltammogram of simultaneous detection of (**a**) lead at a constant concentration of copper; (**b**) copper at a constant concentration of lead at optimized parameters: electrodeposition potential and final potential of −1.5 and +0.25 V respectively, electrodeposition time of 60 s; step potential = 2 mV, step amplitude = 25 mV and frequency = 10 Hz.

### 3.2. Determination of Cu(II) and Pb(II) in Real Samples

The resulting stripping voltammograms for the seed sample are shown in Figure 2.

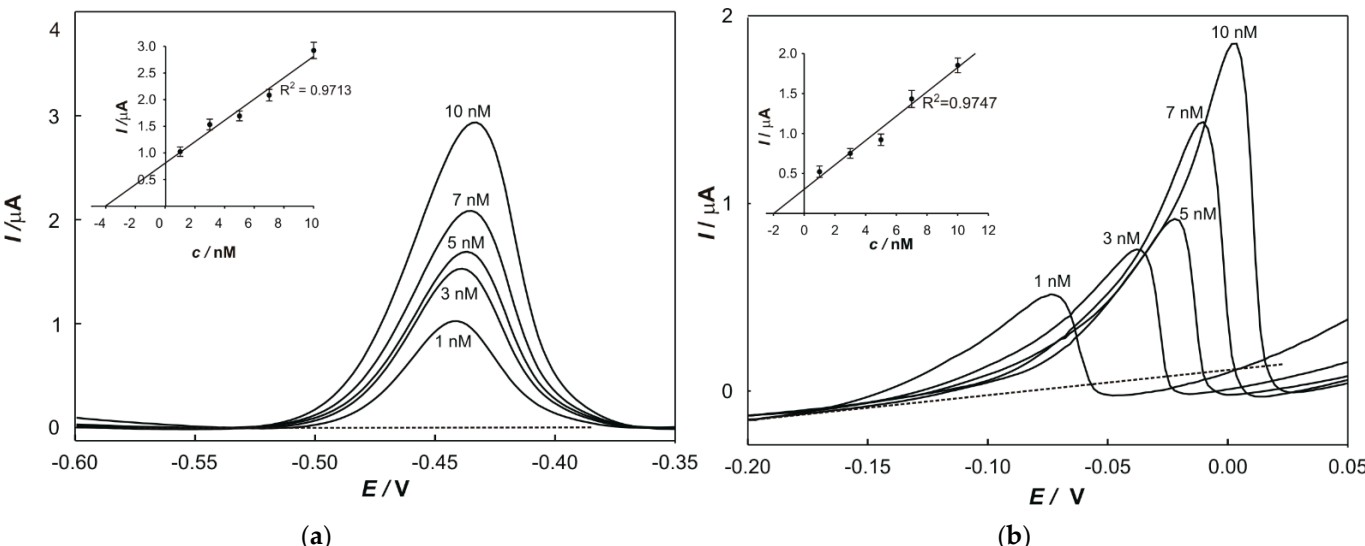

**Figure 2.** SWAS voltammogram recorded for seed sample with (**a**) increasing lead concentration and (**b**) increasing copper concentration. Inset: corresponding calibration curve. Optimized parameters: electrodeposition potential and final potential of −1.5 and +0.25 V, respectively, electrodeposition time of 60 s; step potential = 2 mV, step amplitude = 25 mV and frequency = 10 Hz.

Redox signals for lead were observed at about −0.45 V. The peak potential for copper was −0.07 V, and it gradually shifted toward more negative potentials with an increase in heavy metal concentration.

Obtained voltammograms for the determination of metals in plant samples from S1 and S2, watered with tap water, are shown in Figures 3 and 4.

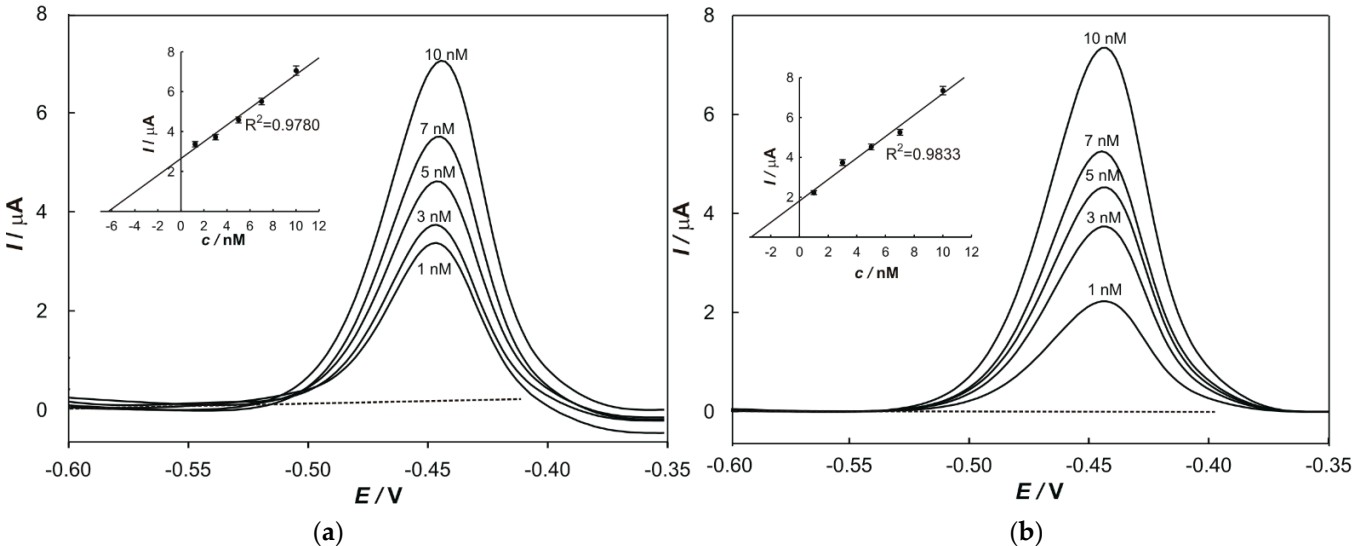

**Figure 3.** SWAS voltammogram of Pb, with the corresponding calibration curve, recorded in plant samples from (**a**) S1; and (**b**) S2; watered with tap water, at optimized parameters: electrodeposition potential and final potential of −1.5 and +0.25 V, respectively, electrodeposition time of 60 s; step potential = 2 mV, step amplitude = 25 mV and frequency = 10 Hz.

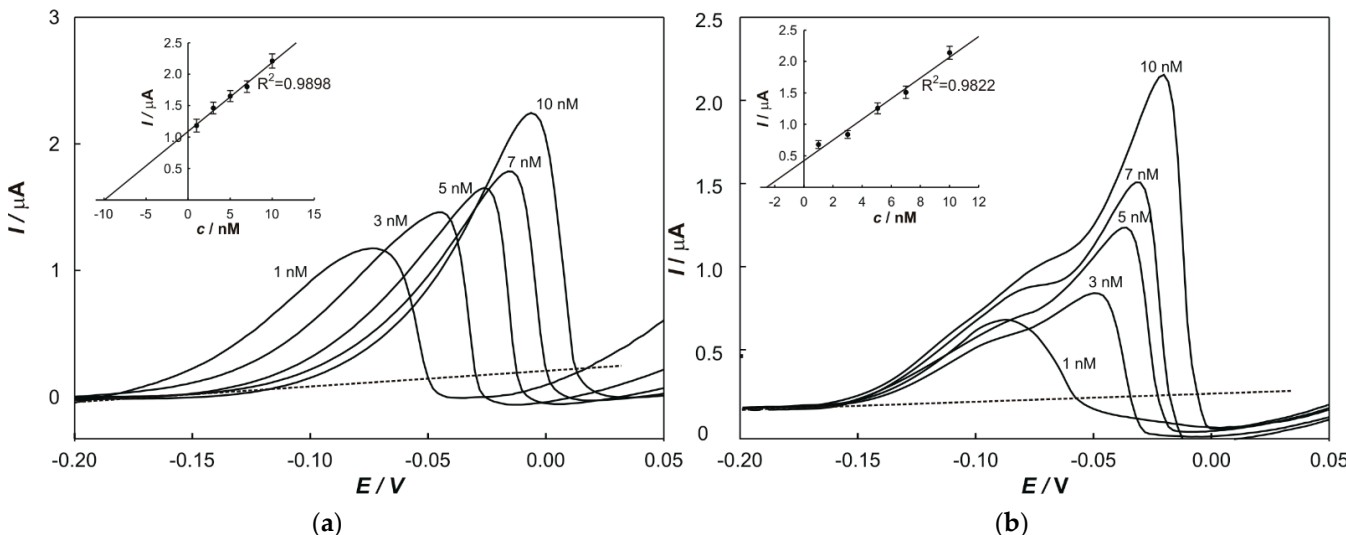

**Figure 4.** SWAS of Cu, with the corresponding calibration curve, recorded in plant samples from (**a**) S1; and (**b**) S2; watered with water, at optimized parameters: electrodeposition potential and final potential of −1.5 and +0.25 V, respectively, electrodeposition time of 60 s; step potential = 2 mV, step amplitude = 25 mV and frequency = 10 Hz.

The voltammograms obtained for the determination of metals in plant samples from S1 and S2, watered with prepared solution (8 mM for Pb and 8 mM for Cu), are shown in Figures 5 and 6, respectively.

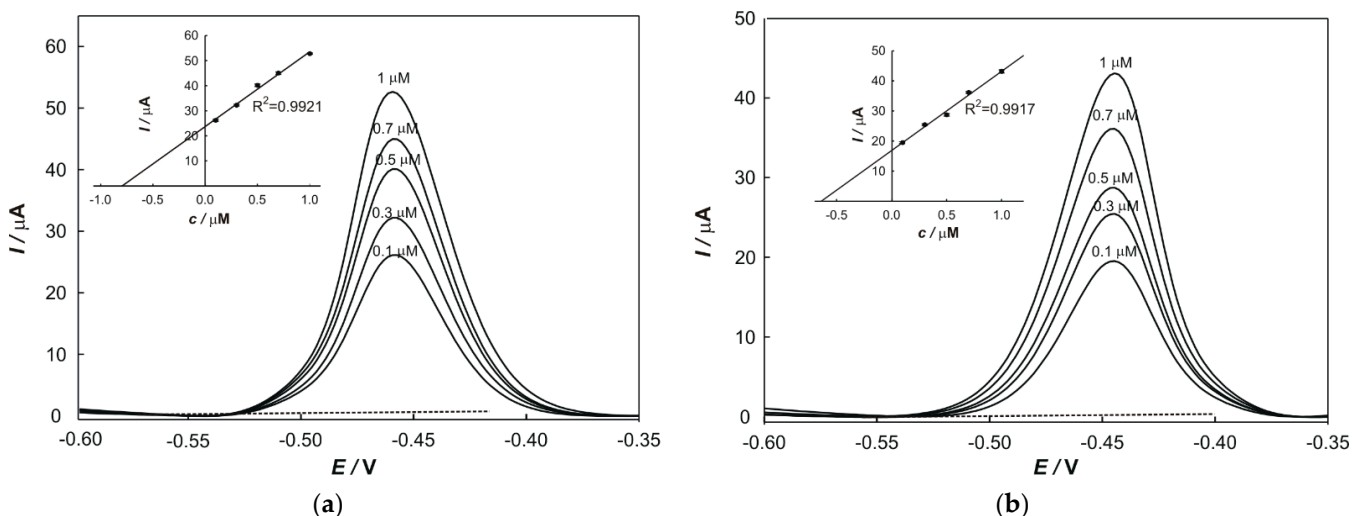

**Figure 5.** SWAS voltammogram of Pb, with the corresponding calibration curve, recorded in plant samples from (**a**) S1; and (**b**) S2; watered with a prepared solution at optimized parameters: electrodeposition potential and final potential of −1.5 and +0.25 V, respectively, electrodeposition time of 60 s; step potential = 2 mV, step amplitude = 25 mV and frequency = 10 Hz.

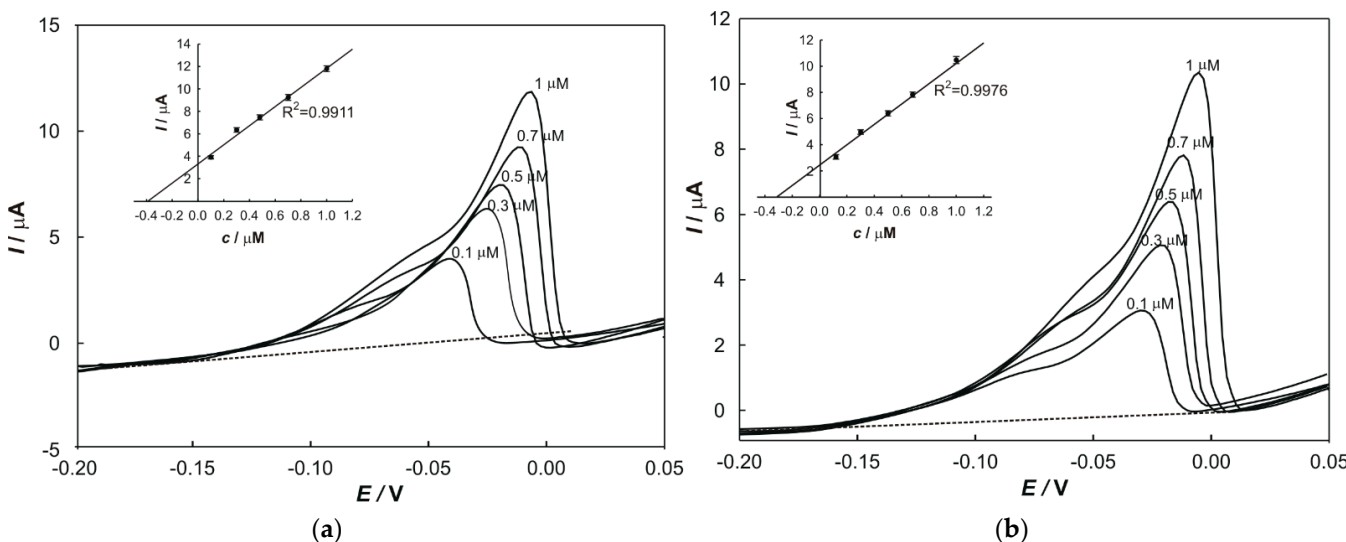

**Figure 6.** SWAS voltammogram of Cu with the corresponding calibration curve, recorded in plant samples from (**a**) S1; and (**b**) S2; watered with a prepared solution at optimized parameters: electrode-position potential and final potential of −1.5 and +0.25 V, respectively, electrodeposition time of 60 s; step potential = 2 mV, step amplitude = 25 mV and frequency = 10 Hz.

After 30 days of exposure, concentrations of Cu in the plant from S1 and S2 watered with tap water were 15.8 ± 1.6 and 4.1 ± 0.6 mg/kg, while the concentrations in the plants from S1 and S2 contaminated with a solution of Pb/Cu were 624.2 ± 23.4 and 498.9 ± 21.0 mg/kg. Unlike for Cu, concentrations of Pb in the plant from S1 and S2 watered with tap water or solution of Pb/Cu were much higher in comparison with unseeded and untreated soil (32.7 ± 2.4 and 17.5 ± 1.6 mg/kg; 4116.2 ± 190.3 and 3314.7 ± 99.4 mg/kg) (Table 1). The detection results were verified by inductively coupled plasma mass spectrometry (ICP-MS). A comparison of voltammetric and ICP-MS results are shown in Table 1.

**Table 1.** The comparison of voltammetric and ICP-MS results for the determination of Pb(II) and Cu(II) in the determined samples.

| Sample (*n* = 5) | ICP | | SWASV | |
|---|---|---|---|---|
| | Content of Pb (mg/kg) | Content of Cu (mg/kg) | Content of Pb (mg/kg) | Content of Cu (mg/kg) |
| S1 | 25.7 ± 0.8 | 29.6 ± 0.8 | 23.8 ± 1.8 | 30.9 ± 1.7 |
| S2 | 10.8 ± 0.4 | 10.8 ± 0.5 | 10.6 ± 0.8 | 11.3 ± 0.9 |
| Seeds Lunaria | 1.6 ± 0.1 | 2.8 ± 0.1 | 2.1 ± 0.3 | 3.1 ± 0.4 |
| S1 + H$_2$O | 25.5 ± 0.8 | 29.8 ± 0.6 | 26.9 ± 1.5 | 29.4 ± 1.7 |
| S1 + Pb/Cu | 2194.4 ± 22.4 | 681.2 ± 15.8 | 2131.8 ± 83.1 | 701.7 ± 36.1 |
| S2 + H$_2$O | 10.8 ± 0.4 | 10.6 ± 0.5 | 11.1 ± 1.1 | 10.6 ± 0.8 |
| S2 + Pb/Cu | 2270.8 ± 39.2 | 715.8 ± 13.3 | 2232.5 ± 75.9 | 708.5 ± 28.0 |
| Plant from S1 + H$_2$O | 28.8 ± 0.8 | 14.7 ± 0.4 | 32.7 ± 2.4 | 15.8 ± 1.6 |
| Plant from S1 + Pb/Cu | 3845.5 ± 52.1 | 598.2 ± 11.3 | 4116.2 ± 190.3 | 624.2 ± 23.4 |
| Plant from S2 + H$_2$O | 16.7 ± 0.3 | 4.0 ± 0.1 | 17.5 ± 1.6 | 4.1 ± 0.6 |
| Plant from S2 + Pb/Cu | 3198.6 ± 47.5 | 501.0 ± 12.5 | 3314.7 ± 99.4 | 498.9 ± 21.0 |

*n*—number of samples in each determination. S1—soil from the Botanical Garden of the Faculty of Science, University of Split. S2—commercially available organic mineral substrate.

In the table, it can be clearly observed that the results obtained from the proposed method are almost the same as those provided using the ICP-MS. This confirms the practical utility of the proposed method for metal ion detection in real samples.

According to the data listed on the product label, S2 contains 12.3 mg/kg of copper and 10.22 mg/kg of lead, which corresponds to the values obtained in these experiments. Higher values of Cu concentrations in S1 could be explained by the fact that it is an agricultural area where vine grapes were planted from where the sample was obtained. Namely, years ago, in the northern part of the botanical garden, autochthonous Mediterranean plant cultures were cultivated (vineyards, olive groves, etc.). In addition, vines were grown near the botanical garden on the slopes of the Marjan forest. It is known that the vines were sprayed with a mixture of copper sulphate, lime and water, popularly known as Bordeaux mixture. In addition, most of the plants in the botanical garden were also repeatedly sprayed with the Bordeaux mixture.

The permissible level of Pb and Cu in agricultural areas in Croatia is 0–50 mg/kg and 0–60 mg/kg [16]. Data in Table 1 show that Pb and Cu levels in the soil samples were under the permissible limit. The values obtained in this work corresponded to those found for the concentrations of Pb and Cu in soil and plants from Brassicaceae families [17].

Concentrations of metals in the samples of plants presented in Table 1 showed that concentrations of Pb in the plant from soils which were watered with tap water are slightly higher than in unseeded and untreated soil, while Cu concentrations did not change significantly.

The amount of lead and copper in plant samples that were watered with a solution of Pb/Cu significantly increased as related to the soil contamination level (S1 and S2). Similar behavior was observed in the article of Herrero et al. [2].

Although the added concentrations of lead and copper were high, the plant did not wither, which suggests that the *Lunaria annua* L. have mechanisms to tolerate the presence of heavy metals in the substrate which they grow in. The results showed that *Lunaria annua* L., according to the criteria for hyperaccumulation [8], is a lead hyperaccumulator and a potential copper accumulator.

According to the literature [1,18], with increasing soil pH, the availability of metals in the soil decreases, so it is more difficult for plants to accept them, i.e., the higher the soil pH, the lower the possibility of metal accumulation. The measured pH values show that soil 1 (pH = 6.35) is poorly acidic to neutral. On the other hand, the purchased organic soil 2 (pH = 8.58) is moderately alkaline soil. So, the obtained results show that the higher the pH, the lower the possibility of metal accumulation. Considering the measured pH values of the investigated soils, this could explain the significantly lower concentrations obtained for copper compared to lead.

## 4. Conclusions

The experiments showed that SWASV could be used successfully to determine metals, particularly copper and lead, in contaminated soils and plant samples. Moreover, such a technique may be a good alternative to spectroscopy due to its simplicity and lower equipment costs.

Obtained concentrations for copper in plants from both types of soil watered with tap water are lower in comparison to the unseeded and untreated soil. Unlike copper, lead concertation in plants from both types of soil watered with tap water were much higher in comparison with unseeded and untreated soil. This confirms the well-known fact that for the bioavailability of copper, the optimal pH should be between 4.5 and 6, while the mobility of lead does not depend on pH value.

The obtained results showed that the higher the pH, the lower the possibility of metal accumulation.

Various species of the Brassicaceae family are known to hyperaccumulate metals. Our results suggest that *Lunaria annua* L. is capable of accumulating toxic amounts of copper and lead. None of the plant samples accumulated copper in concentrations above 1000 mg kg$^{-1}$, meaning the criteria for a hyperaccumulator were not met. However, given

the obtained concentration values, we can conclude that *Lunaria annua* L. is a potential copper accumulator and a heavy metal-tolerant species. In contrast to the obtained values of copper, the values for lead were greater than 1000 mg kg$^{-1}$, indicating that *Lunaria annua* L. is a hyperaccumulator of lead. Hence, the results presented in this paper indicate a possible health risk if the plant material of *Lunaria annua* L. were used for human consumption.

**Author Contributions:** Conceptualization, M.B. (Maša Buljac); methodology, M.B. (Maša Buljac), M.B. (Marijo Buzuk) and J.R.; validation, M.B. (Marijo Buzuk), J.R., D.K. and M.B. (Maša Buljac); formal analysis, N.V. and I.Š.R.; investigation, M.B. (Maša Buljac), M.B. (Marijo Buzuk) and A.M.; resources, M.B. (Maša Buljac) and J.R.; writing—original draft preparation, M.B. (Maša Buljac); writing—review and editing, M.B. (Maša Buljac), M.B. (Marijo Buzuk), J.R. and D.K.; visualization, M.B. (Maša Buljac); supervision, M.B. (Marijo Buzuk) All authors have read and agreed to the published version of the manuscript.

**Funding:** The authors acknowledged the financial support received from the Croatian Science Foundation under the projects BioSMe (Grant IP-2016-06-1316).

**Institutional Review Board Statement:** Not applicable.

**Informed Consent Statement:** Informed consent was obtained from all subjects involved in the study.

**Data Availability Statement:** Not applicable.

**Conflicts of Interest:** The authors declare no conflict of interest.

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
