# Peer review of "Application of the Stripping Voltammetry Method for the Determination of Copper and Lead Hyperaccumulation Potential in Lunaria annua L."

_chemosensors, doi:10.3390/chemosensors10020052_

Round 1

Reviewer 1 Report

In manuscript, Lunaria annua L. was used as a model to investigate the hyperaccumulation potential of copper and lead from contaminated soil with stripping voltammetry. This study is very interesting and practical for environment treatment. The manuscript can be accepted for publication with some revisions:

1. The experimental conditions for electrochemical determination should be optimized, why authors used the mentioned conditions?

2. Some depictions should be corrected, such as 10-5 M

3. For peak current, please use µA in all vertical coordinates;

4. For copper and lead determination, error bars should be added;

5. In Table 1, how many samples were used for each determination, should be provided;

Reviewer 2 Report

Report on the manuscript: “Application of stripping voltammetry method for the determination of copper and lead hyperaccumulation potential in Brassicaceae species”, M. Buljac et al.

Ref. Chemosensors 15441115.

General considerations:

The authors report a study on the determination of the concentration of copper and lead in Lunaria annua L., a vegetal species of the Brassicaceae family using stripping voltammetry at Hg-film over glassy carbon working electrodes. Overall, this is an interesting manuscript that can be published after consideration of several remarks detailed below.

General remarks:

  1. I) The title does not exactly reflect the contents of the manuscript. “Lunaria annua L.” rather than “Brassicaceae species” should be indicated. Only one species of that family is studied.

  1. II) Several experimental aspects should be clarified:

  1. a) What are the sampled plant portions of the plant (leaves, roots, …)?

  1. b) When the plants were harvested (flowering stage, etc.)?

III) Copper was stripped at ca. 0.05 V. At this potential, however, there is opportunity for some oxidation of the Hg film. It would be convenient to add to Fig. 1 the voltammograms of the bare Hg-GCE electrode.

  1. IV) The stripping peak for Pb exhibits in all cases a Gaussian-like profile. However, the stripping peak for Cu shows a more complicated profile (Fig. 4 and 6). In these circumstances, it is unclear if the (apparently measured by the authors) peak current of the major peak is sufficiently representative of Cu2+ concentration in the solution. This problem should be considered. Additionally, the base line used for peak current measurement should be indicated.

  1. V) As is unfortunately frequent in recent literature, the authors express incorrectly the majority of numerical quantities with regard to the number of significant figures. For instance, in page 10, last paragraph, “15.79±1.56 and 4.08±0.64”should be “15.8±1.6 and 4.1±0.6”; “624.16±23.40, etc. Notice that all data in Table 1 are incorrectly expressed. The IUPAC recommendations should be followed.

Minor remarks:

1) The voltammetric parameters and the pH of the water should be indicated in the caption for Figs. 1-6.

Reviewer 3 Report

The manuscript presented describes the application of a mercury film electrode for the determination of lead and copper in plant material. The aim of the study was to demonstrate the usefulness of stripping voltammetry in the study of heavy metal accumulation in plant tissues.

The text of the work requires a thorough revision, clarification of the measurement conditions, description of the samples, and the method of presentation of the measurement results.

Recommendation: major revision

Detailed comments

‘0.25 mm mash’ – mesh

‘The residue was transferred to a 50 ml volumetric flask and diluted to the required volume’ – Residue is an insoluble substance left after decomposition.  What was the point of the flask?

‘Two flower pots (S1 and S2) were watered with 15 mL of tap water and the other two (S1 and S2) with 15 mL of a prepared solution’ – This description is confusing. Does S1 mean ‘soil1’ and S2 ‘soil2’?

‘i.e., initial, electrodeposition and final potential of 1.0, -1.5 and +0.25 V, respectively; electrodeposition and delay time before the potential sweep of 60 and 30 s ‘ – Information provided here should be specified. I assume that the following stages were applied:

  • Electrode was conditioned at potential of 1.0 V for 60 time – Was the mercury film stable at such positive potentials?
  • The target metals were accumulated on the mercury film electrode for 30 s.
  • The positive going square-wave voltammogram was recorded starting from -1.5 V as until a value of 0.25 V was reached.

‘step amplitude = 25 mV, and wave increment = 10 mV’ - Do the authors mean modulation amplitude DE = 25 mV? So what is the quantity defined as ‘wave increment’? Potential step?

‘Stripping analysis consisted of two steps… determined by measuring the generated current’ - This excerpt does not fit into the rest of the chapter. It should possibly be included in the introduction.

‘The values of the peak current obtained at -0.47 V and -0.05 V…’ - According to Figures 2, 4 and 6 in the text, the potential of the copper peak was not constant.

‘…combinations were made with constant Pb2+ (10-7 M) with different additions of Cu2+ (10-6; 10-7; 10-8 M) and constant Cu2+ (10-5 M)...’ What was the point of using such high concentrations of solutions? The expected concentrations of elements in the analyzed solutions were in the range 1e-9 M. When such high concentrations are used, the electrode surface is saturated with respect to the metals mentioned and any changes, even if they might occur, will not be observed because the concentration at the electrode will be constant. The problems resulting from too high concentrations are shown in Figure 6. The copper peaks are asymmetrical, with visible shoulders.

‘empty untreated soil’ – Can it be expressed in any other way?

‘Higher values of Cu concentrations in S1 could be explained by the fact that it is an agricultural area where vine grapes were planted from where the sample was obtained’ - Are the grapes grown in the botanical garden?

‘And it is known that the vine is watered with blue gallop (cupric sulfate anhydrous).’ – Copper-based fungicides are sprinkled on the grapevine instead of watering the plant with copper (II) sulfate.

‘Blue gallop’ -Is this a trade name for a copper preparation used in horticulture?

Round 2

Reviewer 2 Report

The manuscript can be publiushed in its current version.

Reviewer 3 Report

The authors took into account all my comments. I have no other objections to the text.